# Sweet Potato (*Ipomoea batatas* L.) Phenotypes: From Agroindustry to Health Effects

**DOI:** 10.3390/foods11071058

**Published:** 2022-04-06

**Authors:** Alberto A. Escobar-Puentes, Iván Palomo, Lyanne Rodríguez, Eduardo Fuentes, Mónica A. Villegas-Ochoa, Gustavo A. González-Aguilar, Francisco J. Olivas-Aguirre, Abraham Wall-Medrano

**Affiliations:** 1Biomedical Sciences Institute, Autonomous University of Ciudad Juárez, Anillo envolvente del Pronaf y Estocolmo s/n, Ciudad Juárez 32300, Chihuahua, Mexico; abraham.escobar.mcab@gmail.com or; 2Faculty of Medicine and Psychology, Autonomous University of Baja California, Tijuana 22427, Baja California, Mexico; 3Thrombosis Research Center, Department of Clinical Biochemistry and Immunohaematology, Faculty of Health Sciences, University of Talca, Talca 3460000, Chile; ipalomo@utalca.cl (I.P.); lrodriguez@utalca.cl (L.R.); edfuentes@utalca.cl (E.F.); 4Center for Research on Food and Development, Carretera al ejido la Victoria Km 0.6, Hermosillo 83304, Sonora, Mexico; mvillegas@ciad.mx (M.A.V.-O.); gustavo@ciad.mx (G.A.G.-A.); 5Department of Health Sciences, University of Sonora (Campus Cajeme), Blvd. Bordo Nuevo s/n, 7 Ejido Providencia, Cd. Obregón 85199, Sonora, Mexico

**Keywords:** antioxidants, sweet potato, *Ipomoea batatas*, cancer, carotenoids, phenolic compounds

## Abstract

Sweet potato (SP; *Ipomoea batatas* (L.) Lam) is an edible tuber native to America and the sixth most important food crop worldwide. China leads its production in a global market of USD 45 trillion. SP domesticated varieties differ in specific phenotypic/genotypic traits, yet all of them are rich in sugars, slow digestible/resistant starch, vitamins, minerals, bioactive proteins and lipids, carotenoids, polyphenols, ascorbic acid, alkaloids, coumarins, and saponins, in a genotype-dependent manner. Individually or synergistically, SP’s phytochemicals help to prevent many illnesses, including certain types of cancers and cardiovascular disorders. These and other topics, including the production and market diversification of raw SP and its products, and SP’s starch as a functional ingredient, are briefly discussed in this review.

## 1. Introduction

The study of edible roots and tubers (R&T) has attracted the attention of researchers worldwide. Research published to date ranges from their economic and cultural dimensions to their nutritional/functional value as staple foods for certain countries [1,2,3,4]. Among R&T, sweet potato (SP; *I. batatas* (L.) Lam; also known as ‘boniato’, ‘moniato’, ‘caiapo’, ‘kumara’ or ‘kumera’) with its >1600 species, has been a major staple food for certain ancient populations for centuries [1,2]. In fact, archaeobotanical and epigraphic evidence allows us to affirm that SP was and continues to be an ingredient widely used to make different drinks and foods, both sweet and salty, in populations of diverse cultures [3,4].

The genus *Ipomoea* belongs to the *Convolvulaceae* family, and 600–800 species have been identified by cytogenetics [5,6]. Most of them exhibit health-promoting bioactivities, such as those related to their phytochemical profile: anti-inflammatory (*I. cairica*), anti-constipation (*I. digitata*), analgesic (*I. stans*), antidiabetic and hypotensive (*I. aquatica*, *I. batatas*), hemostatic and vasoconstrictor (*I. tricolor*), psychotomimetic (*I. muelleri*, *I. violacea*) and anticancer (*I. horsfalliae*, *I. turpethum*) activities [7]. Sweet potato (SP; *I. batatas* (L.) Lam), contains a wide range of nutrients and xenobiotic phytochemicals with antioxidant, anti-nyctalopia/xerophthalmia, hepatoprotective/spasmolytic, anticoagulant/anti-HIV antibacterial, and antidiabetic potential. Particularly, specific anticancer bioactives (e.g., phenolic acids, carotenoids, and peptides) present in the aerial (leaves, steams, talks) and non-aerial (storage roots) parts of SP suggest phenotype/varietal-specific benefits [8,9,10,11,12]. It is noteworthy that certain phenotypic traits of SP genotypes are closely related to their functional/nutraceutical value: the peel and flesh (central parenchyma) pigmentation, going from white-creamy to dark purple (Figure 1) is related to their phenolic and carotenoid content [13,14,15,16].

However, food processing and preservation [13,14,17,18,19] and the gastrointestinal fate of its phytochemicals [20,21,22] may hinder the health-promoting potential of SP. The aim of this narrative review is to provide an update on SP’s botany/molecular phylogeny, agroindustry, and product commercialization/technological diversification, as well as the nutritional/functional value of SP’s major genotypes (by flesh color) and certain health effects (cancer chemoprevention and cardiovascular health promotion). Certain physiological considerations to ensure SP’s health benefits are discussed shortly.

### 1.1. Botany and Molecular Phylogeny

SP is a member of the bindweed or morning-glory (*Convolvulaceae*) family that groups ~60 genera and >1650 species. This perennial herbaceous vine is a dicotyledonous initially described by Linnaeus in 1753 as *Convolvulus batatas* and further reclassified by Jean-Baptiste Lamarck in 1791 within the genus *Ipomoea*, based on its stigma shape and pollen grain surface. The systematic botany, based on the phenotypical features of a typical SP plant, was previously reported by Huamán [23], who described that storage roots—the commercial edible fraction mistakenly known as a tuber—differ in size, color of rind, skin (peel), flesh (central parenchyma) and shape [24]. Currently, one out of three *Ipomoea* accessions deposited in recognized gene banks (e.g., GenBank^®^) is *I. batatas* [25]; most of them are commercialized in regional markets.

The origin, timing, and geographic location(s) of cultivated SP have been disentangled recently with the advent of molecular phylogenetics [11,25]. Domesticated SP initially diverged from two non-edible ancestors: the wild SP (polyploid; “pencil-shaped” roots); and *I. trifida* (diploid; no root-forming). These evolved spontaneously to other tetraploids or hexaploid [26,27,28] progenitors with much thicker tuberous storage roots. Their subsequent clonal propagation, under a pre-existing environmental trait and complex (yet unknown) growing factors, resulted in wild edible storage roots that preceded the domestic cultivation of current SP genotypes. Nowadays, it is widely accepted that SP are native to South-Central America (domesticated >5000 years ago) and were introduced to Spain and Europe (by Columbus), Africa, India, Southeast Asia, and the Philippines (by Portuguese explorers) between the 14th and 16th century, and that its intensive cultivation began ~630 years ago in Asia and in the 1960s in Africa [29]. It is noteworthy, that the Quechua and Polynesian names for SP (‘kuumala’ and ‘kumara/cumal’), and further DNA genotyping support the Peruvian origin and human transfer of the Polynesian SP [30,31].

A successful breeding program for SP implies a careful selection of germplasms (genetic diversity) and the systematic evaluation of phenotypic traits (ideotypes) sensitive to environmental stressors and crop agronomic performance indicators [32,33,34]. It is noteworthy that genetic breeding programs aiming to produce new SP varieties remain challenged since it is an allohexaploid crop [5,26,27,28] with a large chromosome number (2n = 6x = 90), a complex sporophytic self-and cross-incompatibility and a high degree of genomic duplication [35,36]. Besides genetic diversity, SP varieties phenotypically differ in flesh/skin colors, size, shape, texture, and taste of the storage root, all intrinsically related to a variety-specific activation/regulation of key biosynthetic routes, such as the methylerythritol-4-phosphate (for carotenoids) and phenylpropanoid (for anthocyanins) pathways [24]. New hexaploidy cultigens with agronomical and nutritional profiles have been produced by genetic engineering. SP phenotypes with higher Fe/Zn bioaccessibility, better carotenoid (IbOr), anthocyanin (IbMYB1, IbDFR), or starch (GBSSI) profiles as well as climate-resilient (e.g., ‘Radiance’), and ‘dual-purpose’ varieties, are some examples [14,25,37,38,39,40,41].

### 1.2. Agroindustry and Phytochemistry

Roots and tubers (R&T) are the third most important food crop after cereals and legumes. R&T with the highest production value are cassava (*Manihot esculenta*), potato (*Solanum tuberosum*), yam (*Dioscorea alata*), and SP [42]. SP can grow in many soils and environmental conditions and demand few agricultural resources, and grow rapidly (3–5 months), depending on the genotype-by-environment interaction [14,43,44]. SP does not tolerate cold weather, but neutral/alkaline and humus-free soils are essential for obtaining high-quality storage roots [45]. SP is not as perishable as climacteric fruits and vegetables, yet post-harvest losses due to physical, physiological (malformed shape), and pathological (microbial spoilage) damage can be important, affecting the SP’s shelf life and consumer preference for SP [46,47]. Recommendations for a successful SP harvest are beyond the scope and purpose of this review, but they have been discussed in depth by other authors [13,14].

The global harvested area (Ha × 10^6^) of SP between 1961 and 2010 fell from 13.4 to 8.1, yet a positive production (0.98 to 1.10 tons × 10^8^) and yield (hg/Ha × 10^4^) trend was recently reported [46]. However, the global SP production did not change much between 1999–2019 (Table 1 and Appendix A). According to FAO [47], the average harvested area (Ha × 10^6^), annual production (tons × 10^8^), and yield (hg/Ha × 10^4^) were 9.7, 1.5, and 8.9 in 1999, 7.8, 1.0, and 10.0 in 2009, and 7.8, 0.9, and 11.0 in 2019, respectively (Table 1a). According to FAO, SP is currently cultivated in 112 countries (Appendix A), China and Nigeria contributing to 69.4%/ 87.4% (1999), 55.5%/ 76.1% (2009), and 52.7%/ 61.1% (2019) of the global harvest area (Ha)/production (tons) of this tuber, with the current market value at nearly USD 45 trillion. It is noteworthy that China’s and Nigeria’s crop yields (Hg/Ha) were not as good as those of other small producers (≤14, 615 Ha), such as Senegal, Australia, Egypt, and the Cook Islands in the same years (Table 1b). A reduction in harvest area was also documented in China and Africa between 1961–2010 [47].

Global crop yields (hg/Ha) had a slight decline in 1999 (16.5%), 2009 (12.8%), 2019 (11.8%), a fact closely associated with a country (United States of America (USA), China (Ch)), continent (Africa (Af), America (Am), Asia (As)), and worldwide (W) reduction in the total harvested area (Appendix A; R^2^ = 0.84).

Nevertheless, China (21.2–21.5–21.9), Asia (19.6–19.4–20.2) and Africa to a lesser extent (6.3–5.5–5.3) maintained their yields despite reducing their harvested area and net production (Table 1 and Appendix A). On the other hand, in Latin American countries (8.8–9.7–14.9) and the United States of America (16.5–22.5–24.4), crop yields increased significantly even with a lower harvestable area, compared to Asian countries. As documented by other authors [48,49], between 1961–2010, Brazil, Cuba, and Haiti were major producers (5.2 ± 2.0 × 10^4^ Ha/3.9 ± 2.3 × 10^5^ tons; Appendix A). Success stories in novel agricultural practices (e.g., poly-cross nurseries) have been documented in other Latin American countries, such as Peru and Bolivia, where SP is a part of their culinary history [50,51,52].

Moreover, root and tuber crops that provide both food and energy with a low environmental cost (such as SP) are particularly beneficial for the economic progress of small farmers—a mandatory action to achieve the Global Sustainable Development Goals (SDGs) 1.0, 2.0, and 7.0 [43]. However, many specific market reports clearly indicate an increasing trend in the importation and regional production of SP, jointly driven by the international consumer demand, novel retailing, and distribution channels, but mostly a climbing market share pushed by China as the leading producer [46,47,48,49]. 

#### 1.2.1. Market Insights

SP is an attractive crop for alleviating many of the world’s nutritional problems, such as chronic malnutrition (energy), and hidden hunger (micro deficiencies) [31,53,54]. For centuries, SP has been considered a major staple food in certain middle and low-income countries, unfairly labeled as a “poor person’s”, “orphan”, “subsistence” or “famine relief” crop [49,51], despite other health benefits [8,9,12] not related to its energetic value (~390 kcal·100 g^−1^). In fact, SP produces more edible energy/Ha than any other R&T crop (see caloric content in Appendix A), and in African countries, the daily consumption of SP and other R&T represents one-third of the daily energy intake [55]. This “food security” demand for fresh SP has contributed to a seemingly stable current regional market [13,14] that drove a global market growth (million USD) from 48,629 in 2019 to 58,470 in 2027 (Compound Annual Growth Rate (CAGR) = 2.1%), with the Asia-Pacific region holding the largest market share (China 80–83%) due to their own domestic consumption and total imports from other countries (e.g., the USA and Uganda) to fulfill that need [41].

The current food and beverage industry of SP-based products is segmented by application, form, type, and by end-use, besides the obvious geographic segmentation. According to Innova Market Insights [56], global launches of SP-based products increased at a CAGR of 21% between 2015 and 2018 and were: baby foods (14%), cakes, pastries, frozen vegetables, ready-to-eat meals, and confectionary, the latter because of the unique color of certain varieties. Moreover, people’s awareness of the health risks associated with wheat gluten prompted a new market niche for SP chips and fries that will grow at a 5.1% CAGR between 2022 and 2030 [57]. It is noteworthy, however, that most of these food developments are based on orange-fleshed SPs while white-creamy or purple ones are on their way.

Lastly, the diversification of SP-based ingredients or products by processing technologies will improve their market penetration even more so, generating new revenues [43]. Some alternative food technologies to home cooking include freezing (cubes, chunks, strips), canning (juices, purees, jams, jellies), thermal and non-thermal dehydration (flours, flakes), frying and baking (chips, strips), and microbial fermentation (beverages, vinegar, pickle, curd, yogurt) technologies or their sub-fractionation to obtain starches, sugars, alcohol, natural (food) colorants and/or semi-purified extracts [14]. As expected, such food developments partially depend on the cultivar genotype, since each one has its own physicochemical [58] and sensorial [59] signature. A detailed description of these processing technologies (advantages and limitations) has been reported by others [13,14].

#### 1.2.2. Nutritional Value

The raw chemical composition of representative white, yellow, orange and purple-fleshed SP is reported in Table 2. 

As indicated by other authors [7,8,9,10,12,13,14,15,16,17], SPs mainly consist of carbohydrates (sugars + starch) and protein, crude fiber, fat, and ash in graded order. Although food processing [10,18,19,20,21] and genetic improvements [5,35,36,37,43,51,52] modify the content of specific components, the overall ratio often remains intact. Moreover, like many other R&Ts, SPs are rich in essential minerals (e.g., Mg, Mn, Fe, P, Zn, Cu Ca) and vitamins, such as α/β- carotene, lutein, vitamin B1, B2, B6, pantothenic acid, niacin, biotin ascorbate, and tocols [10,60,61].

The total/specific content of carotenoids (provitamin A) in SP varies substantially by plant part, varietal (genotype/phenotype), and food processing. In general, SP is recognized as an excellent source of provitamin A (β > α carotenes, 829-43200 IU) [20,21,32,60]. Orange-fleshed SP (OFSP; Figure 1) stands as the best SP source of β-carotene and total carotenoids, and certain varietals, such as Tomlins, Owori, Bechoff, Menya, and Westby varietals rank higher (20–364 µg/g DW) than other recognized β-carotene sources such as carrots (43.5–88.4 µg·g^−1^ dw) or mango (10.9–12.1 µg·g^−1^ dw) [61]. This phenomenon supports the relationship between organoleptic properties, such as the visual and sensory acceptance by consumers as well as the position of multiple researchers who have recommended SP consumption as an appropriate strategy to combat global problems, such as food safety and deficit syndromes such as xerophthalmia or night blindness [62].

The *Nutrient Rich Foods Index* (NRF9.3) has been proposed as a valid and comprehensive index to select individual foods based on their nutrient density (richness), allowing the identification of foods rich in essential nutrients (protein, fiber, vitamins A, C, E, calcium, iron, magnesium, and potassium) while limiting those unhealthy ones, such as saturated fats, added sugars and sodium [60]. OFSP´s NRF9.3, canned and/or mashed (925.2), raw (298.9), boiled (166.9), and chips (147.2) rank higher than more than two hundred vegetables (e.g., carrot= 102.0), three hundred fruits (e.g., apple = 47.1), sixty beans, nuts, and seeds (~23.1) and SP leaves/fries/confections (<24.4). Moreover, when considering nutrient density (as NRF9.3) vs. nutrient affordability (as NRF9.3.price^−1^·100 kcal^−1^; higher (Q1) to lower (Q2) cost), raw OFSP (NDP No. 11507) also stands out as a plant food with the highest nutritional value at a lower lowest cost (Figure 2).

It is noteworthy, however, that major contributors to such a favorable NRF9.3 come from OFSP’s provitamin A content and the lower simple (sugars)-to-complex (starch and dietary fiber) carbohydrate ratio (Table 2). In this sense, starch is the major complex carbohydrate for SP and its content differs among varieties (58–73% dry matter) [63] while the total content of sucrose, fructose, and glucose—although presented in a lesser amount when compared to starch—is responsible for the sweet taste of most SP varieties. However, SP’s peel or flesh (central parenchyma) are not rich sources of crude dietary fiber (Table 2), although they are of slow-fermentable, digestible, and resistant starch, which from the point of view of functional nutrition, is very convenient for the formulation of food for people with glycemic and gastrointestinal disorders [64,65].

The nutritional and nutraceutical properties of natural sources of complex carbohydrates are highly influenced by their GI digestion patterns. Particularly, natural starches are classified according to their hydrolyzing rate under simulated GI conditions as rapid (RDS, 0–20 min), slow (SDS; 21–120 min) and resistant (RS) starch fractions, and the latter is considered mostly fermentable by resident colonic microflora (prebiotic effect) in vivo or under simulated colonic conditions [21,63]. RS fermentation produces metabolites (e.g., short-chain fatty acids) related to the prevention of carcinogenesis, the improvement of insulin resistance and the prevention of diseases related to metabolic syndrome [8,10,41].

Total starch and RS content in SP differ upon phenotypes, pre/post-harvest practices, and geographic location, among other factors. SP’s RS content varies between 3–68 g·100 g^−1^ [66,67,68,69], being purple (6.2–38), red, OFSP (15–25) and white (6–10) [9,70,71]. In contrast, some authors have reported that the RS content does not vary (24–25%) significantly in starch in different varieties of SP (yellow, white, and purple) [72]; these differences in SP’s RS content are explained by the differences in their amylose/amylopectin ratio. *Quantitative Structure-Activity Relationship* (QSAR) refers to the process by which a chemical structure (e.g., a given SP phytochemical) is quantitatively correlated with a well-defined biological (e.g., molecular docking with a cell receptor or enzyme), chemical (e.g., the molecular affinity of one molecule to another) and technological (e.g., Maillard reaction) reactivity and so, mild varietal-specific SP’s amylose/amylopectin ratio (Figure 3) may impact both the technological food properties and ultimate bioactivity and chemical behavior along the GI tract.

The prebiotic activity of SP flours or semi-purified SP starches has been demonstrated in vitro on *Bifidobacteria animalis*, *Lactobacillus acidophilus*, and *Lactobacillus casei* [9,73]. Several authors have proposed the incorporation of new technologies/treatments to increase the RS content and the functional value of SP. Furthermore, from a technological standpoint, prolonged post-harvest storage [74], autoclaving and enzymatic debranching [75], retrogradation and acetylation [65], annealing [66,72], or heat-moistening [72] is used to improve the RS content in SP flour and/or starch. Moreover, chemically-induced esterification (e.g., by succination) increases substantially (up to ten times) SP’s RS content [67,68]. 

Lastly, the protein contribution of SP is within the range of 1.2–6.2 g/100 g of dry weight (DW). However, the quality of this protein is high as it contains several essential amino acids (mg/g^−1^ DW), such as leucine (1.2–2.4), isoleucine (0.7–1.5), lysine (1.1–2.2), methionine (0.2–0.3), phenylalanine (0.9–1.8), threonine (0.08), tryptophan (0.8–1.7) and valine (1.1–2.1), which are necessary for the proper functioning of the human body [76]. As if this were not enough, SP protein hydrolysates have been shown to have an antioxidant capacity that helps prevent oxidative DNA damage [77].

#### 1.2.3. Functional Value

In addition to the nutritional benefits previously mentioned, SPs contain a wide range of phytochemicals with antioxidant capacity (flavonoids), anti-nyctalopia/xerophthalmia (carotenoids), hepatoprotective/spasmolytic (scopoletin), and antibacterial (friedelin), among other health benefits extensively documented by others [7,8,12,78]. Although the regulation of these events is surely associated with synergistic activities due to the phytochemicals contained, carotenoids and polyphenolic compounds have received particular interest from the scientific community, derived from their abundance and diversity in SP. Conventionally, it has been conservatively reported that purple SP varieties possess the highest content phenolic content, followed by OFSP and white varieties (Table 3). It should be noted that content, as well as the diversity of bioactive compounds, is closely related to the color of its flesh [78,79,80]; however, some phenolic species are common among the varieties.

Nowadays, it is possible to discriminate the metabolic profiles of several SP cultivars through chemometrics, where main phytochemical differences could be targeted to activate/disactivate metabolic pathways either naturally [81,82,83] or by genetic inbreeding [5,28,35,36,51]. Particularly, high-through output chromatographic platforms have revealed common and variety-specific phenolic profiles. The body of evidence indicates that SPs have a high content of hydroxycinnamic acids with quinic and caffeic acid derivates commonly present in various genotypes. Among the most abundant hydroxycinnamic derivatives reported are chlorogenic acid, ferulic acid-o-hexoside, feruoyl quinic acid and 3,5-di-caffeoyl-quinic, while the aglycone forms (e.g., caffeic or ferulic acid) are less abundant [81,82]. As for flavonoids, Wang et al. [81] reported that quercetin is one of the major components belonging to flavan-3-ols in purple SP, while Kampferol is abundant in all the varieties examined. These species, as well as catechin, luteolin, chrysoeriol, and hesperetin, have been consistently reported in other investigations [83]. These structures are not found in isolation but are associated with glucose or galactose with O-glycoside bonds. The glycosylation patterns, in essence, represent a challenge for the identification not only of these but also of species such as anthocyanins.

Anthocyanins are particularly reported in purple SP varieties. This makes sense since they are the molecules responsible for providing the characteristic color to both the peel and the flesh. The main anthocyanins reported in purple SP are cyanidin-3-O-glucoside and peonidin 3-O-glucoside [79,80,81] in the form of monoacetylation and diacetylation. It is necessary to point out that the analysis of transcripts in the phenylpropanoid pathway has revealed high participation of the genes *IbC4H*, *IbCHS*, *IbCHI*, *IbF3H*, *IbDFR*, *IbANS*, *and IbUGT*, which do not occur equally in white, yellow, or orange varieties [81]. Moreover, as previously mentioned, carotenoids are major non-phenolic antioxidants compounds, and some varieties of SP also have a great diversity of carotenoid pigments and are distributed as follows (µgEβ-carotene/100 g): OFSP (180), yellow (16), white (4.5), and purple (2.9) [84,85]. Phenolic compounds + carotenoids + ascorbate synergistically contribute to SP’s antioxidant capacity (mg of Trolox equivalents (TE).100 g^−1^) in a varietal-dependent manner [86]: Purple (17.2–17.9), yellow (9.45–13.45), OFSP (6.98–11.8), and white (3.17–17.6). 

Saponins are natural compounds made up of aglycone and oligosaccharide chains, which have active surface properties that have been poorly explored in SP. The extraction, isolation, and identification of saponins is a challenge for the scientific community given the diversity of possible glycosides as well as the absence of comparative standards. However, one of the most relevant studies in the field showed that SP can contain about 200 mg of saponins per 100 g (dw). The saponins reported after chemical hydrolysis were oleanolic acid-3-O-[b-d-glucopyranosyl-(1→2)-b-d-galactopyranosyl-(1→2)-b-d-glucuronopyranosyl]-28-O-b-d-glucopyranoside (sandrosaponin IX) y oleanolic acid-3-O-[b-d-galactopyranosyl-(1→3)-b-d-glucuronopyranosyl]-28-O-b-d-glucopyranoside; these molecules also possess high antioxidant capacity through both antiradical and reducing properties [87]. Lastly, triterpenoids (e.g., bohemeryl acetates, friedelin, β-amyrin) and coumarins (e.g., aesculetin, scopoletin, and umbelliferone) are minor SP phytochemicals with antioxidant, antimicrobial, antinociceptive, anticoagulant, anti-HIV replication, hepatoprotective, spasmolytic, and anti-acetylcholinesterase activity [12,88]. 

### 1.3. Health Effects and Metabolic Fate of SP’s Phytochemicals

Once the diversity of nutrient and non-nutrient compounds has been examined, it is not surprising to find multiple reviews in the literature highlighting the biological activities attributable to SP in the maintenance of optimal nutritional states, and in the prevention of various diseases [8,9,10,11,12,13,14,15,16]. SP’s antioxidant, antimicrobial, anti-diabetic, anti-cancer, anti-inflammatory, hepatoprotective, neuroprotective, anti-obesity, and GI-health-promoting properties have been extensively reviewed [88,89,90]. However, the subsequent section focuses on relevant information on the anticancer and anti-cardiovascular disease (CVD), pathologies with the highest morbidity/mortality rates worldwide, in which SP can contribute to conventional clinical treatments as alternative adjuvants.

#### 1.3.1. SP and Cancer

Plant bioactives exert many benefits in cancer chemoprevention; cyto/genotoxicity, cell cycle arrest, pro-apoptosis, intracellular signaling, immunomodulation, and anti-angiogenesis are probably the most studied mechanisms. Specifically, a robust body of evidence indicates that certain antioxidant phytochemicals, such as phenolic compounds, carotenoids, ascorbate, and antioxidant dietary fiber and RS, can halt the progression of certain types of cancer cells in vitro and ex vivo, although their effectiveness under clinical conditions remains uncertain. Personalized nutrition for cancer patients demands a continuous search for newer sources of phytochemicals to be used in complementary and alternative medicine. Several studies carried out in recent years [91,92,93,94,95,96,97,98,99,100,101,102,103,104,105,106,107,108,109] have reported multiple control points determining the process of initiation, promotion, or the spread of cancer (Table 4).

In the early stages, the cellular integrity, and in particular the genetic material, can be preserved by various compounds, such as trypsin inhibitors, anthocyanins, protein hydrolysates, or hydroxycinnamic acids present in the various varieties of SP. Damage to genetic material caused by reactive oxygen species (^●^OH o H_2_O_2_) o UV/gamma-irradiation can be decreased (if not avoided) after exposure to cells with functional compounds that have a high antioxidant capacity [91]. Additionally, it has been shown that the trypsin inhibitor present in the SP variety Tainong 57 can increase the expression of the protein p53, a nuclear protein known as “the guardian of the genome” due to its role in limiting abnormal cell formation, thus preserving the integrity of the genetic material [92]. 

If cell integrity is affected, this cell becomes abnormal and must multiply to promote a cancerous process. Cyclins are molecular mediators of the cell cycle; in normal conditions, they require binding with kinases to promote the different phases of cell division. Phenolic extracts of the SP variety Whatle/Loretan can decrease the expression of these proteins and limit complexing with their respective kinases [93]. According to the event and in the case of SP phytochemicals, Huang et al., have demonstrated the potential of trypsin inhibitors present in the Tainong 57 variety to limit cell division after arrest in the early phases (G_1_ phase) of altered cells [92].

As might be expected, the regulation of signaling pathways is a common and functional mechanism for the modulation of different tumors. The anthocyanins present in SP have been shown to be effective in negatively regulating the signaling pathway of β-catenin, a protein widely recognized for acting as a permanent coactivator of events, such as cell proliferation and differentiation [94,95]. In addition to the role of anthocyanins, phytosterols such as β-Sitosterol-d-glucoside play important roles in the regulation of additional pathways. β-Sitosterol-d-glucoside has been shown to be efficient in negatively regulating the PI3K/AKT/mTOR signaling pathway, a key pathway in processes, such as cell proliferation, apoptosis, metabolism, and angiogenesis [96,97]. The adverse systemic effects associated with the cancer process largely depend on tumor formation, its ability to survive, nutrient acquisition, and location, among other issues. Phenomena such as angiogenesis have been related to the capacity present in transformed cells to produce chemical mediators that promote vascularization and, therefore, the growth of tumor cells and their dissemination throughout the body (metastasis). Therefore, bioactives combinations with anti-angiogenic capacity seem essential in limiting processes such as the promotion and spread of cancer [98].

Chen et al. [99] demonstrated the ability of SP polyphenols to reduce the expression of the Vascular Endothelial Growth Factor (VEGF165) in a dose-dependent manner. Moreover, it has been shown that the glycoprotein SPG-56 present in the SP variety Zhongshu-1 can modulate the expression of essential proteins in cell attachment and adhesion. Dysregulation in the production of proteins, such as claudins or occludins (essential for the formation of tight junctions between cells), has been reported under in vitro conditions [100]. Beyond cancer promotion or progression stages, cell cytotoxicity on its own deserves attention. In all stages, the induction of cell death by apoptosis is a key tool to stop the number of viable cells in a programmed way. This event has already been reported for SP polyphenols. Particularly, the anthocyanin fractions from SP P40, O’Henry, NC Japanese as well as Bhu Krishna seem to be effective modulators in cell models [101,102]. 

The multi-target nature of SP’s phytochemicals helps to tackle cancer at many stages; however, future research on this matter should consider the SP varietal richness and plant part [88], gastrointestinal bioavailability [20,21,22,37,50], and pharmacokinetics of a given SP’s bioactivity to guarantee the effects observed in vitro/ex vivo conditions.

#### 1.3.2. SP and Cardiovascular Diseases (CVD)

CVD are the leading cause of worldwide adult mortality. Prevalent cases of total CVD nearly doubled from 271 (CI_95%_ 257–285) to 523 (CI_95%_ 497–550) million deaths and 17.7 to 34.4 million disability-adjusted life years (DALYs) between 1990–2019 [110]. Since CVD and other non-communicable chronic diseases are closely related to lifestyle factors (e.g., unhealthy diet and sedentarism), it is necessary to promote the healthy intake of fruits, nuts, seeds, beans, vegetables, whole grains, and R&T [111], including the aerial/non-aerial parts of SP plant [88]. Numerous investigations indicate that the dietary intake of flavonoids (e.g., quercetin) from plant foods such as purple SP, can reduce the risk for CVD [112] while SP’s tannins, flavonoids, alkaloids reducing sugars, anthraquinones, and cardiac glycosides reduces serum creatinine and lactate-dehydrogenase activity, favoring cardiovascular health [88]. 

Consuming SP leaves reduces the risk for CVD by synergistically reducing lipid peroxidation and DNA damage, and regulating blood glucose, insulin, and lipid levels [113,114]. Such metabolic effects are partially explained by the 1: 2 ratio of linoleic/α-linolenic fatty acids [115], compounds that can protect the cardiovascular system from excessive inflammation and oxidative damage [116]. Moreover, Zhao et al. [114] showed that flavones from an SP leaf powder decreased total cholesterol and triglyceride levels in a dose-dependent manner while its insoluble dietary fiber increased fecal bile acids and cholesterol, reducing serum cholesterol levels [117]. In support of this, a randomized controlled clinical study carried out on 58 humans showed a decrease in circulating cholesterol (7 mg/dL) and triglycerides (2 mg/dL) after the consumption of 132 g of white SP as a meal replacement [118]. Moreover, it has been demonstrated in hamsters that consuming SP leaves increases the presence of favorable biomarkers to reduce the risks for CVD [118] by inducing vascular (aortic) relaxation [119] mediated by nitric oxide (NO) as an inhibitor, in the presence of N ω -nitro-l-arginine (NOLA), an inhibitor nitric oxide synthase (NOS), or by eliminating it from the endothelium [120]. As for SP root, <3 kDa hydrolyzed peptides (VSAIW, AIWGA, FVIKP, VVMPSTF, and FHDPMLR) from sporomin A and B, display a strong anti-ACE (angiotensin-converting enzyme) activity [88,121], while lactic acid bacteria (LAB)-based fermentation of white (Murasaki), orange (Evangeline) and purple (NIC-413)-fleshed SP varieties increases their anti-ACE/antioxidant activity [122]. Additional evidence on the cardioprotective effects of extracts of SP and/or its pure phytochemicals previously identified by chromatographic techniques is summarized in Table 5.

In conclusion, this evidence suggests that several SP bioactives (leaves/root) may individually and synergistically prevent CVD by exerting many cardioprotective mechanisms. Further investigations on the associated molecular events are needed to support the epidemiological and in vivo and in vitro evidence discussed above.

## 2. Final Remarks

SP is a common staple food in certain middle/low-income countries with great nutritional and functional potential. Its health benefits are not limited to its nutrients but also to other xenobiotics, including resistant starch [64,65], antioxidants [19,78,79,80,81,82,83,84,111], terpenoids [12,88], phytosterols [96,97], bioactive peptides [88,121,122] and many other phytochemicals that modulate key metabolic processes, reducing the odds for chronic illnesses including, yet not restricted to certain types of cancer [92,94,96,100,102,103,104,105,106,107,108,109,110] and CVD. Further research should focus on understanding the physiological mechanisms and metabolic biotransformation of raw/processed SP’s bioactives whose intrinsic functionalities (e.g., anti-inflammatory, antioxidant, enzyme-inhibitory activity) target multiple target organs [88,89,90].

## Figures and Tables

**Figure 1 foods-11-01058-f001:**
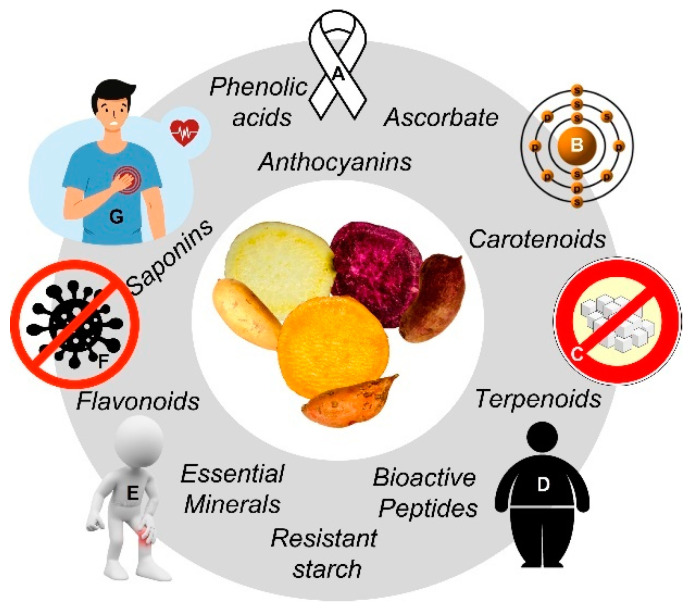
Sweet potato (SP; *Ipomoea batatas* L.) group of phytochemicals with associated health-promoting effects. Preventive actions (clockwise): Immunocompromise (**A**), prooxidant (**B**), diabetes (**C**), adiposity (**D**), inflammatory (**E**), infection (**F**), cardiovascular (**G**) diseases/metabolic rearrangements. Source: The authors (CC (by/nc/sa)-licensed clip art).

**Figure 2 foods-11-01058-f002:**
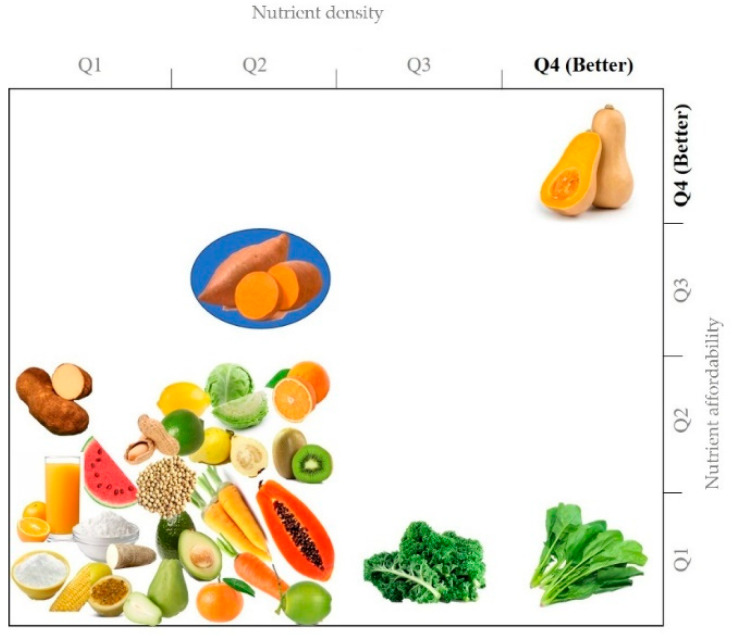
Nutrient density and affordability of SP-based foods as compared to common fruits and vegetables. Nutrient density (*Nutrient Rich Foods index* (NRF9.3), see references [60,61] for calculation details) vs. nutrient affordability (NRF9.3.price^−1^·100 kcal^−1^; higher (Q1) to lower (Q2) cost) plot.

**Figure 3 foods-11-01058-f003:**
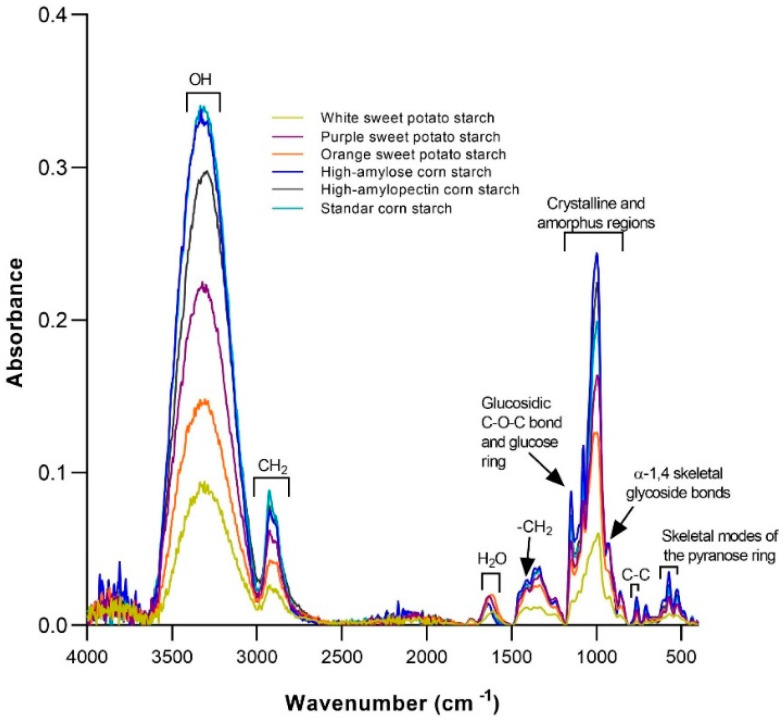
FTIR spectra from white, orange, and purple sweet potato (SP) starches as compared to corn starches (CS: high-amylose, high-amylopectin, and normal). Functional chemical groups in SP are quite like those found in CS, although with a variety-specific intensity. Absorption signals at 1048 and 1022 cm^−1^ are associated with crystalline and amorphous starch regions, while the intense peak at 1000 cm^−1^ can be attributed to amylose and amylopectin molecules. Source: The authors.

**Table 1 foods-11-01058-t001:** Agricultural performance of sweet potatoes in two decades ^1^.

(a)	Global	China		Nigeria	
1999					
Area (Ha)	9,749,117	5,945,975		817,000	
Production (tons)	147,214,978	126,143,701		2,354,000	
Yield (Hg/Ha)	89,239 ± 63,996	212,150		43,673	
2009					
Area (Ha)	7,848,391	3,253,056		1,100,000	
Production (tons)	96,424,362	70,040,593		3,300,000	
Yield (Hg/Ha)	100,364 ± 73,753	215,307		30,000	
2019					
Area (Ha)	7,769,851	2,373,737		1,717,659	
Production (tons)	91,820,732	51,992,156		4,145,488	
Yield (Hg/Ha)	110,770 ± 82,415	219,031		24,135	
(b)	1999	2009	2019	ΔYield/y	R^2^
Senegal	87,469	400,000	385,997	15,396	0.69
Australia	275,000	250,797	363,976	4805	0.59
Egypt	240,545	285,425	320,537	4197	0.99
Cook Islands	263,478	266,667	291,667	1502	0.86

^1^ Source: FAO statistical databases [47]. Agronomic performance indicators for large (a) and small (b) SP producers, average yield change/year (ΔYield/y), and linear trend (R^2^).

**Table 2 foods-11-01058-t002:** Nutrient composition of SP of different flesh colors ^1^.

Component	White	Yellow	Orange	Purple
Total carbohydrates	85.3–87.3	81.3–85.7	83.1–87.0	84.5–85.0
Digestible starch	54.6–64.1	51.2–61.1	42.3–60.0	53.4–54.8
Sucrose	5.0–12.9	7.7–11.6	4.7–16.5	5.8–8.1
Protein	4.1–5.8	5.1–5.9	4.3–6.2	5.4–5.8
Resistant starch	2.5–3.7	1.6–4.3	0.6–3.8	1.8–2.7
Ash	2.3–3.4	2.6–2.8	3.3–4.5	1.5–2.9
Crude fiber	1.6–2.6	1.3–1.4	1.9–3.3	1.1–1.5
Fructose	0.5–4.5	0.8–4.3	0.9–6.6	1.9–2.4
Glucose	0.6–4.8	0.9–1.3	1.0–6.5	1.8–2.3
Fat	1.3–1.7	1.8–2.1	1.3–2.2	1.3–1.8

^1^ Range (min–max) content (g·100 g^−1^, dry weight basis). Data source: [7,8,9,10,12,13,14,15,16,17].

**Table 3 foods-11-01058-t003:** Antioxidant phytochemicals reported in SP (*Ipomoea batatas* L.) ^1^.

Parameter	White	Yellow	Orange	Purple
Total phenols (mg GAE)	1.4–2.5	3.3–3.5	2.9–4.6	11.5–12.3
Flavonoids (mg QE)	5.8–12.2	27.3–29.6	14.6–29.6	76.2–84.4
DPPH (mg TE)	3.2–17.6	9.5–13.5	7.0–11.8	17.2–17.9
Anthocyanins (mg Cy3GE)	-	-	-	1.4–1.6
Carotenoids (mg)	4.5	16.0	180.0	2.9

^1^ Range content (g·100 g^−1^, dry weight basis). Gallic acid (GAE), Quercetin (QE), Trolox (TE) and Cyanidin 3-O-glucoside (Cy3GE) equivalents, negligible content (-). Data source: [78,79,80].

**Table 4 foods-11-01058-t004:** Bioactive compounds in SP and their role against cancer.

Variety	Phytochemical	Mechanism	Action
Initiation			
Tainong 57	Trypsin inhibitor	DNA damage reparation	↑ P53 leukemic cells
--	Polyphenols	↓ ROS	↓ Oxidative damage induced by H_2_O_2_ in HepG2 cells.
Mixuan No. 1	Protein hydrolysate	↓ ROS	↑ antioxidant activity, ↓ oxidative damage to DNA
Ayamurasaki	Anthocyanins	↓ROS	↓ Oxidative damage induced by radiation in thymocytes
Tainong 57	Trypsin inhibitor	Cell cycle arrest	Phase G1 arrest
TU-155	Polyphenols	Cell cycle arrest	↓ciclin D1, A y E, ↑ Cip1/p21
Promotion			
NING No. 1	Polysaccharides	Anti-inflammatory	↓ IL-1β, IL-6 y TNF-α
TNG 73	Anthocyanins	Anti-inflammatory	↓ activation of NF-*κβ* in RAW 264.7 cells induced by LPS
--	Caffeic acid and derivates	Inhibition in cell proliferation	β-catenin and Tcf-4 pathway suppression
Progression			
Bhu Krishna	Anthocyanins	Cell death induction	Apoptosis—↑ caspases
Diverse	Anthocyanins	Cell death induction	↑ caspase 3 in colonic cells
--	Polyphenols	Angiogenesis inhibition	↓ VEGF165 in a dose-dependent manner
--	BSG	Invasion inhibition	PI3K-Akt signaling pathway suppression
Zhongshu-1	SPG-56 Glycoprotein	Invasion inhibition	Regulation in the expression of proteins (MMP2, MMP9, VEGF, ocludin, and claudin) related with metastasis.
TNG 73	Anthocyanins	Invasion inhibition	Cell migration suppression (MCF-7 cells)

Non specified (--), β-Sitosterol-d-glucoside (BSG). Data source: [92,93,94,95,96,100,101,102,103,104,105,106,107,108,109].

**Table 5 foods-11-01058-t005:** SP phytochemicals in cardiovascular diseases (CVD).

Phytochemical	Mechanism	Action
Heart		
Anthocyanins	↓ Malondialdehyde	Antioxidant ↓ Lipid peroxidation
Flavonoids/ anthocyanin	Vasodilation induction/ ↓ endothelin—1	Antihypertensive
Tannins/saponins/ Flavonoids/terpenoids	↓ Creatine kinase ↓ Lactate dehydrogenase	Prevention in ischemic damage
Vascular		
Aqueous extracts	↑ Telomerase activity preventing cell senescence	Prevention of coronary artery disease
Anthocyanins	Inhibition of PDGF receptor-β	Regulation of platelet aggregation
Chlorogenic acid	ACE Inhibition	Antihypertensive
Anthocyanins/ethanolic extract	↓ VCAM	Prevention of atherosclerosis
SP leaves	Elongate arterial occlusion time	Prevention of thrombotic events
Purple SP extract	↓ cyclooxygenase-2, ↓ inducible nitric oxide synthase ↓ tumor necrosis factor-α	↓Inflammation
Brain and Kidney		
Anthocyanins	↑ BDNF	Neuroprotection after ischemic stroke
Flavonoids/ acetylated anthocyanins	Blocking VEGFR2/ROS/NLRP3 signaling	↓ Kidney damage

Angiotensin-converting enzyme (ACE), brain-Derived Nuclear Factor (BDNF), NLR family pyrin domain containing 3 (NLRP3), reactive oxygen species (ROS), platelet-derived growth factor (PDGF), sweet potato (SP), vascular cell adhesion molecule (VCAM), vascular endothelial growth factors receptor 2 (VEGFR2). Data source: [123,124,125,126,127,128,129,130,131,132,133,134].

## Data Availability

Not applicable.

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
