# Peer review of "Sweet Potato (Ipomoea batatas L.) Phenotypes: From Agroindustry to Health Effects"

_foods, 2022, doi:10.3390/foods11071058_

Round 1

Reviewer 1 Report

Line 30: please mention the sources according to your statements.

Line 65: no need to mention the different names of SP, since already mentioned above.

Line 71: “previously reported Huaman”. Please add “by”

Line 73: please mention the sources.

Section 1.2 Agroindustry; in this section, you have mentioned that the production of SP in some countries are increasing/decreasing, do you have any idea/explanation for this? It would be nice if you can provide your analysis this phenomenon.

Section 1.2.1 Market insight: it would be nice if you can provide the data in graph/table.

Line 222, it isnoteworthy, no space in the sentences.

Please describe  what QSAR is referring to,  in the discussion.

Line 271, please compile the information in Table.

Line 358, “o irradiation” it seems typo, please correct it.

Author Response

Authors response-Reviewer 1 (see PDF file)

Reviewer 2 Report

I have now reviewed the article: Sweet potato (Ipomoea batatas L.) phenotypes: From agroindustry to health effects. The work is interesting and I have the following issues in order to improve the manuscript:

  1. Figure 1 is not clear at all and needs redesign or redoing to show the  chemical constituents with their structures
  2. Table 2 requires clarification to avoid the reader confusion: Carbohydrates components should be made clearer to calculate the total amount correctly. Also the monosaccharides, fructose, glucose and sucrose. I suppose the values are intact sugars as determined but what about the hydrolysis of sucrose during the assays?
  3. Figure 2 is really not clear, nor the purpose of it. The A part can be separated if necessary.
  4. Figure 3, FTIR spectra seem to be interesting and should be made a separate figure so make it clearer. I do not know what is the importance of the right part?
  5. Table 3: Flavonoids should also be estimated as GAE for comparison. The authors might keep the values in reference to Quercetin. 
  6. A Table should be made for SP and cardiovascular diseases (CVD) in a similar way to that for SP and Cancer.
  7. I do not agree that Sweet potato tuberous staple diet in the whole world. Therefore I think the authors should investigate a comparison of SP data with the most popular different Potato species (which are much cheaper and available) throughout the world. What are the real benefits of SP over other species?
  8. Some statements in the final remarks (conclusions) are not backed by evidence and therefore should be deleted.

Author Response

See attached response letter

Reviewer 3 Report

Sweet potato, Ipomoea batatas L., is an important staple food in several regions of the world. The scientific literature clearly shows that sweet potato is a good source of nutrients and bioactive compounds with essential positive effects on human nutrition and health, hence the relevance of this review.

This manuscript provides a botanical description and molecular phylogeny of sweet potato. In addition, it presents and briefly discusses its nutritional composition in bioactive compounds and impacts on human health. Furthermore, it provides information related to the agro-industry and commercialization of products of this species. The authors selected the most relevant data from the literature and presented them to reflect their particular importance to the scientific community.

The manuscript is easy to read and it is well written. However, please consider the following comments:

  1. The paper focuses on a topic of interest. However, structurally, the manuscript has 2 points: Introduction and conclusions. The authors should consider reorganizing the manuscript's structure. I think that this work would be better structured and more enlightening, if the authors organized the introduction into several independent topics, ending with Final Remarks.
  2. In the introduction describe the objectives more clearly.
  3. Despite the fact that the manuscript is well written, throughout the text there are extremely long sentences that need to be revised, such as the ones I mention as an example:

Line 105-110 “Lastly, dual-purpose SP varieties (DPSVs) have been developed for both human (storage roots) and livestock (vines) consumption [14, 39], while new faster-maturing cold-resistant varieties such as “Radiance”, a red-skinned, orange-fleshed Canadian genotype developed by the Vineland Research & Innovation Centre (VRIC), will be fulfilling the seasonal supplying gaps of orange-fleshed SP for the international market [40, 41].”

or

line 114-119 “SP can grow in many soil & environmental conditions, demands few agricultural resources, and grows rapidly (3-5 115 months) depending on the genotype-by-environment interaction [14, 43, 44]; however, the plant does not tolerate cold weather or frost at harvest and adequate drainage, sowing in soils without clay, neutral-to mild alkaline and humus-free, is essential for obtaining high quality storage roots [45].”

  1. Place the bibliographic references in tables 1, 2 and 3.
  2. In table 1, clarify the meaning of “a)” and “b)”  in the table subtitle. Also, clarify the meaning of ΔYield / y e R2 (it is not clear why this value appears in the table)
  3. In Figure 3, clarify the meaning of “a)” and “b)” in the Figure subtitle.

Author Response

Authors response-Reviewer 2 (see PDF file)
